# Sketch2Diagram: Generating Vector Diagrams from Hand-Drawn Sketches

**Itsumi Saito**[*,†]**, Haruto Yoshida**[*]**, Keisuke Sakaguchi**[*,†]

[*]Tohoku University, [†]RIKEN AIP

`itsumi.saito@tohoku.ac.jp`

## Abstract

We address the challenge of automatically generating high-quality vector diagrams from hand-drawn sketches. Vector diagrams are essential for communicating complex ideas across various fields, offering flexibility and scalability. While recent research has progressed in generating diagrams from text descriptions, converting hand-drawn sketches into vector diagrams remains largely unexplored due to the lack of suitable datasets. To address this gap, we introduce SKETIkZ, a dataset comprising 3,231 pairs of hand-drawn sketches and thier corresponding TikZ codes as well as reference diagrams. Our evaluations reveal the limitations of state-of-the-art vision and language models (VLMs), positioning SKETIkZ as a key benchmark for future research in sketch-to-diagram conversion. Along with SKETIkZ, we present IMGTIkZ, an image-to-TikZ model that integrates a 6.7B parameter code-specialized open-source large language model (LLM) with a pretrained vision encoder. Despite its relatively compact size, IMGTIkZ performs comparably to GPT-4o. This success is driven by using our two data augmentation techniques and a multi-candidate inference strategy. Our findings open promising directions for future research in sketch-to-diagram conversion and broader image-to-code generation tasks. SKETIkZ is publicly available.[1]

## 1 Introduction

Diagrams serve as powerful visual tools widely adopted across academic and professional domains to communicate complex ideas effectively. They play a crucial role in clear communication and knowledge transfer by distilling complex information into more accessible visual formats. Vector graphics have become the standard medium for creating high-quality diagrams, primarily due to their inherent scalability and precision. The ability to resize and modify vector diagrams without degrading quality makes them especially valuable in academic and professional settings. These characteristics enable researchers and professionals to adapt diagrams seamlessly across different presentation formats and requirements, enhancing both the clarity and versatility of scientific communication. While established tools and languages such as TikZ and Graphviz are popular for creating high-quality vector graphics, they often require significant manual effort and specialized expertise. Recent developments in large language models (LLMs), such as GPT-4o, have triggered a growing interest in automating the generation of vector graphic diagrams from textual descriptions (Belouadi et al., 2023; Zala et al., 2023; Zou et al., 2024). This emerging research area holds significant potential to enhance the efficiency of diagram creation and improve accessibility to high-quality visualizations. Despite the significant advancements in text-to-code generation, generating diagrams *from sketches* remains largely unexplored. Sketch-based input often provides a more intuitive and user-friendly way to express visual ideas (Figure 1). This approach leverages the inherent human ability to quickly and effectively communicate complex visual information through simple drawings. A primary reason for the limited research in this area is the lack of publicly available datasets that pair hand-drawn sketches with their corresponding codes. Such datasets are essential for training and evaluating models that translate sketch-based input into structured diagram code.

To address this gap, we introduce SKETIkZ, a new dataset designed for benchmarking *sketch-to-diagram generation*. SKETIkZ comprises 3,231 pairs of hand-drawn sketches and their corresponding TikZ codes. The sketches were created using several tools commonly employed in real-world scenarios: paper, whiteboards, and tablets. This diverse collection provides a valuable resource for

---

[1]`https://sketikz.github.io/`

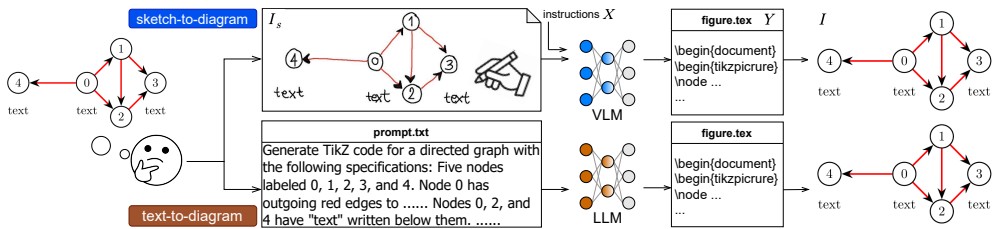

Figure 1: Overview of *sketch-to-diagram*. We consider scenarios where users hand-draw diagrams that they want to create. Sketch-to-diagram models (e.g., VLM) take these sketches $I_s$ and predefined instructions $X$ and then generate code $Y$ for producing vector graphics. $Y$ is subsequently rendered into generated image $I$. The process of *text-to-diagram* is also provided for comparison.

advancing research in automated diagram generation from sketches. SKETIᴋZ aims to facilitate the development of models capable of generating high-quality diagrams from hand-drawn inputs for real-world applications. We also developed IMGTIᴋZ, a Vision-Language Model (VLM) specifically designed for this task. Our model combines three components: an open-source LLM specialized in code generation, a vision encoder, and an adapter. This combination is intended to create a model capable of efficiently converting sketches into TiᴋZ code. We evaluated the effectiveness of two strategies: expanding our dataset through data augmentation and employing an inference strategy that generates multiple candidates and selects the best one. From the results, IMGTIᴋZ performed comparably to GPT-4o in subjective evaluations despite having a relatively small model size of 6.7B parameters. However, both IMGTIᴋZ and the latest state-of-the-art models still struggle to accurately generate code that captures all elements and layouts of sketches, indicating the potential for further advances. We aim for our dataset and findings to drive future research and development in this field. Our contributions are summarized as follows:

- We introduce SKETIᴋZ: A new dataset containing 3,231 pairs of hand-drawn sketches and their corresponding TiᴋZ codes, addressing the lack of real-world data for sketch-to-vector diagram conversion.

- We develop IMGTIᴋZ: An image-to-TiᴋZ model that combines a 6.7B parameter code-specialized LLM with a pre-trained vision encoder, achieving performance comparable to larger models despite its modest size.

- We empirically demonstrate the effectiveness of two types of data augmentation and a multi-candidate inference strategy.

## 2 RELATED WORK

**Vision and language models**   With advancements in LLMs, significant progress has been made in constructing VLMs that interpret images and generate text. A promising approach integrates vision encoders like CLIP (Radford et al., 2021) with LLMs via adapter modules. This method has demonstrated promising results (Liu et al., 2023; Dai et al., 2023; Ye et al., 2023; Zhu et al., 2023; Li et al., 2024; Wang et al., 2024), efficiently creating VLMs that leverage the extensive knowledge base of pre-trained models. In this study, along the same line as these approaches, we build a VLM to generate TiᴋZ code from images.

**Image-to-code generation**   While VLMs are primarily designed to generate natural language outputs, such as answering questions and describing images, research on generating code for image rendering—such as HTML, LaTeX, or SVG—has emerged as a valuable application. For instance, recent studies have introduced models capable of generating LaTeX code from screenshots of mathematical formulas or handwritten images (Deng et al., 2016; Gervais et al., 2024), HTML code from web page screenshots (Soselia et al., 2023; Si et al., 2024; Laurençon et al., 2024; Gui et al., 2024), and SVG code from icon images (Rodriguez et al., 2023). While LaTeX code generation and TikZ code generation are similar in terms of code output, our research tackles significantly more complex problems than previous formula-to-LaTeX conversion studies. It involves much longer output sequences (739 tokens on average compared to 65 tokens in prior work) and requires an understanding

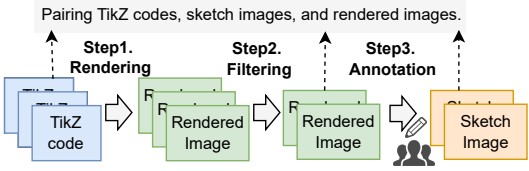

Figure 2: Dataset construction process.

Table 1: Sketch Tool Usage Statistics.

| Tool | Number | Proportion |
|---|---|---|
| Paper | 2,545 | 78.8% |
| Whiteboard | 346 | 10.7% |
| Tablet | 340 | 10.5% |
| All | 3,231 | 100% |

of two-dimensional layouts. We introduce three key advances to handle this increased complexity: code-specialized VLM, two data augmentation strategies, and multi-candidate generation.

**Diagram understanding**   Understanding diagrams has been an important and long-standing research topic, including question answering (Kembhavi et al., 2016; Lu et al., 2023; Wang et al.), caption generation (Hsu et al., 2021; Singh et al., 2023; Huang et al., 2023), and generating descriptions (Hu et al., 2023; Bhushan & Lee, 2022; Bhushan et al., 2024). Recent research proposed benchmark datasets to assess not only the understanding of diagram images but also the direct comprehension of vector graphics code (Zou et al., 2024; Qiu et al., 2024). This expanding research area reflects the growing interest in understanding vector graphics diagrams.

**Diagram generation**   Ellis et al. (2017) proposed a model generating TikZ code for primitive geometric sketches, focusing on circles, rectangles, and lines without text. We extend the approach to handle real-world diagrams with unrestricted shapes and text. Furthermore, our dataset reflects realistic environments by including sketch images from various sources such as paper, whiteboards, and tablets. Recent work has explored real-world diagram generation from text (Belouadi et al., 2023; Zala et al., 2023). Belouadi et al. (2023) proposed a method for generating TikZ code to render diagram images from caption text. Generating diagrams through code synthesis provides better controllability and editability than pixel-based image generation methods, while enabling LLM integration. Belouadi et al. (2023) also highlights the challenge of image-to-diagram generation, which remains limited due to the scarcity of paired image-code data. Concurrent work by Belouadi et al. (2024) addresses the task of generating diagrams from images, which is closely related to our task. However, their evaluation of sketch-based generation is limited to a small dataset, which lacks corresponding TikZ code and thus cannot be used for image-to-code training. Our dataset provides the largest and most diverse sketch-to-diagram dataset with TikZ code, captured under real-world conditions. We also contribute novel data augmentation methods and multi-candidate generation strategies, providing new insights for future research directions in this field.

## 3 DATASET AND TASK

### 3.1 TASK DEFINITION

We introduce a *sketch-to-diagram* task (Figure 1), where the input consists of a sketch image of a diagram $I_s$ and a language instruction $X$, and the output is a sequence of TikZ code $Y$. Then generated TikZ code $Y$ are compiled to render the diagram image $I$.

### 3.2 DATASET CONSTRUCTION

We constructed our dataset in three steps: rendering, filtering, and sketch annotation (Figure 2).

**Step 1: Rendering diagrams from TikZ code**   We first rendered diagrams from TikZ code in the DaTikZ (Belouadi et al., 2023) by using pdflatex. We then paired the rendered reference diagrams $I_r$ with the corresponding TikZ code $Y_r$. We refer to the rendered diagrams as the reference images.

**Step 2: Diagram classification and filtering**   Diagrams can be classified into various categories, as demonstrated by ACL-Fig (Karishma et al., 2023) with its 19-category dataset. For our sketch-to-diagram task, we focused on diagrams composed of geometric shapes and arrows, excluding those primarily based on numerical data. We specifically targeted diagrams categorized as Tree, Graph, Architecture Diagram, Neural Networks, and Venn Diagram according to ACL-Fig labels. We chose these categories because sketch-to-diagram generation is particularly effective for visually oriented diagrams. These diagrams often involve complex combinations of shapes and interconnections, making manual creation time-consuming and precise linguistic instructions challenging. Using an

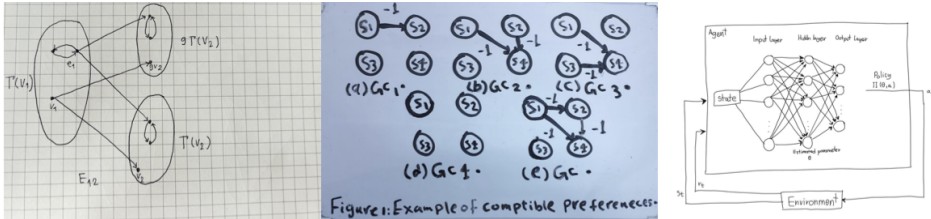

Figure 3: Examples of sketch images. Left: paper, Center: whiteboard, Right: tablet.

Table 2: Datasets used for training IMGTIkZ.

| No | Name | Input | Output | Size | Stage1 | Stage2 |
|----|------|-------|--------|------|--------|--------|
| 1 | arXiv figure | Figure or table image | OCR text | 1.2M | ✓ | |
| 2 | arXiv figure | Figure or table image | Caption text | 1.1M | ✓ | |
| 3 | LLaVA-Pretrain[2] | Multi-domain image | Caption text | 558K | ✓ | |
| 4 | SKETIkZ | Diagram sketch image | TikZ source code | 2.6K | | ✓ |
| 5 | RenderTikZ | Diagram image | TikZ source code | 155K | | ✓ |
| 6 | AugTikZ | Diagram image | TikZ source code | 556K | | ✓ |
| 7 | ImgAugTikZ | Noised Diagram image | TikZ source code | 714K | | ✓ |
| 8 | DaTikZ-v2[3] | Diagram image | TikZ source code | 46K | | ✓ |

image classification model trained on the ACL-fig dataset (details in Appendix E), we extracted and sampled 4,000 diagram images from our targeted categories for annotation. We present the detailed breakdown of categories in Table 10 and Figure 8 in Appendix E.

**Step 3: Sketch data collection** Twenty-eight annotators created sketch images $I_s$ on the basis of filtered reference images $I_r$. Annotators used black pens primarily, with red, blue, and green for colored elements, excluding complex diagrams and ignoring color filling. Regarding the sketching tools, annotators freely selected an available option from paper, whiteboard, or tablet the basis of their respective environments. Table 1 shows the distribution of sketches by tool, with the paper being the most common. Figure 3 illustrates examples of sketch images drawn using each tool. The dataset includes diverse sketches mimicking real-world scenarios, with paper and whiteboard sketches showing varied lighting and backgrounds. We aligned sketches $I_s$ with corresponding TikZ codes $Y_r$ and reference images $I_r$, creating a dataset of 2,585 training, 323 validation, and 323 test samples. More examples are shown in Appendix F

## 4 IMGTIkZ: VISION-LANGUAGE MODEL FOR IMAGE-TO-TIkZ GENERATION

### 4.1 MODEL STRUCTURE

We developed IMGTIkZ, a VLM specifically designed for this task using the model architecture of LLaVA 1.5 (Liu et al., 2023). The model architecture comprises three key components: a code-specialized LLM, a vision encoder, and an adapter, illustrated in Figure 4 (a). The model inputs a diagram image and generates a corresponding TikZ code. We employed the same architecture as LLaVA 1.5 for the adapter module - a simple two-layer multi-layer perceptron (MLP). While the original LLaVA 1.5 uses a language model for natural language generation, we replaced it with a 6.7B instruction-tuned DeepSeek Coder (Guo et al., 2024) for code generation. For vision encoder, we used SigLIP model Zhai et al. (2023). We trained our model in two stages: first updating only the adapter parameters, then training both adapter and LoRA (Hu et al., 2021) parameters added to the LLM. The LLM and vision encoder parameters remained frozen throughout training. For more detailed information about the model hyperparameters, refer to Appendix B and Table 8.

### 4.2 TRAINING DATA

**Datasets used in stage 1 training** In stage 1 training, we incorporated arXiv figure data (No. 1 and 2 in Table 2) in addition to LLaVA-pretrain data (No. 3). This arXiv figure dataset was created by extracting figures, tables, and captions from arXiv paper PDFs in arXiv bulk dataset

---

[2]https://huggingface.co/datasets/liuhaotian/LLaVA-Pretrain
[3]https://huggingface.co/datasets/nllg/datikz-v2

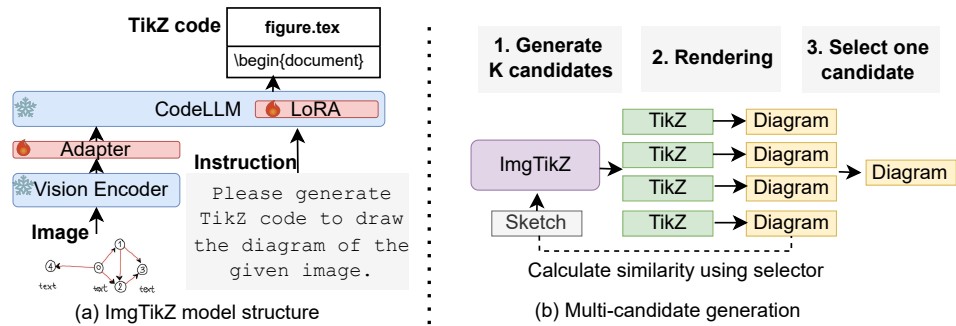

Figure 4: IMGTI*k*Z model structure (a) and multi-candidate generation process for inference (b).

using PDFFigure2.0.[4] We also used Google Cloud Vision API[5] to extract text from these images. The arXiv data served two purposes: (1) generating optical character recognition (OCR) text from images to improve text recognition and (2) generating captions from diagram images to enhance diagram image understanding.

**Datasets used in stage 2 training**    In the second stage of training, we focused on enhancing the model's ability to generate Ti*k*Z code. Given the limited size of the SKETI*k*Z dataset alone, we supplemented our training data by creating pairs of rendered diagram images and Ti*k*Z code collected from the arXiv source file in bulk data, which is referred to as RenderTi*k*Z (No. 5). We implemented two data augmentation techniques to increase diagram and image variations. First, we generated Ti*k*Z code using GPT-3.5 to increase the variety of diagrams, referred to as AugTi*k*Z (No. 6). Second, we applied an image augmentation technique, referred to as ImgAugTi*k*Z (No. 7), to simulate common sketch image noise such as background interference, lighting variations, and rotation. In addition to these augmentation techniques, we also used existing pairs of Ti*k*Z code and images (No. 8), excluding data with arXiv IDs that overlap with our collected dataset.

**Data augmentation for increasing diagram variations**    While we collected approximately 916K original TikZ codes from arXiv sources, many failed to be compiled during RenderTi*k*Z creation. We used GPT-3.5 to fix these compilation errors with a prompt such as "Please modify the code to make it compilable." To increase diagram variety, we instructed GPT-3.5 to modify the original diagram into a different diagram, producing altered versions of the original diagrams. These augmentation techniques resulted in 556K AugTi*k*Z data samples. Previous data augmentation for VLMs used other VLMs to generate instruction-response pairs from images, which was costly due to image processing. Instead, we generate data efficiently by modifying only Ti*k*Z code using text-based LLMs. This approach could be applied to various image-to-code tasks. More details are in Appendix G.1.

**Data augmentation for increasing image variations** Hand-drawn sketch diagrams inherently contain more image noise than rendered images. This noise can appear as background interference or lighting variations when capturing sketches from paper or whiteboards. Furthermore, handwritten text and lines often exhibit significant distortions, and diagrams are frequently stored with angular rotations. To address these issues, we applied multiple image augmentation techniques to RenderTi*k*Z and AugTi*k*Z datasets, such as synthesizing notebook backgrounds, adding Gaussian noise, varying brightness and contrast, and introducing distortion. Figure 5 illustrates an example of the augmented image. This augmentation approach is general-purpose and can be applied to various sketch-to-diagram tasks. More details are in Appendix G.2.

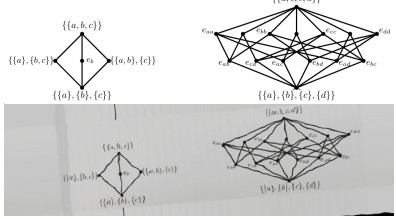

Figure 5: Example of ImgAugTi*k*Z. Top: original image, bottom: augmented image.

---

[4]https://github.com/allenai/pdffigures2
[5]https://cloud.google.com/vision/docs?hl=en

### 4.3 INFERENCE

We implemented two inference methods: iterative generation and multi-candidate generation. In the paper, we refer to them as IMGTi*k*Z-IG and IMGTi*k*Z-MCG, respectively.

**Iterative generation**  Iterative generation produces one candidate per test sample, regenerating upon compilation failure until success. We set a maximum number of generation attempts $M$ to limit this process. This method is straightforward and can be considered a baseline approach.

**Multi-candidate generation**  Multi-candidate generation creates $K$ candidates simultaneously, selecting the best one (Figure 4 (b)) using a selector model. In our study, we generate multiple Ti*k*Z codes and render them as images. The selector selects the best candidate by maximizing the similarity between the input sketch image $I_s$ and the generated diagram image $I$. As general vision encoders cannot accurately measure diagram similarity, we propose D-SigLIP (Diagram-Specialized SigLIP) as our selector. D-SigLIP adds a trainable linear layer to a pre-trained SigLIP model, and we fine-tune only this layer through contrastive learning (Chen et al., 2020) with noise-augmented diagram pairs from RenderTi*k*Z and AugTi*k*Z. More details are in Appendix C. To calculate the similarity score, we computed the cosine similarity between the embedding vectors obtained by inputting the sketch image $I_s$ and the generated diagram image $I$ into D-SigLIP.

Our task requires generating lengthy code sequences (averaging 739 tokens), making producing error-free code in a single-generation attempt challenging. Furthermore, since the model training is based on next-token prediction loss for code sequences, metrics related to image quality are not explicitly considered during code generation. The multi-candidate generation and selection strategy allows us to evaluate these metrics after code generation, which could not be considered during the training phase. While similar approaches have been proposed for text inference and coding tasks (Brown et al., 2024), our work is the first to use image similarity for candidate selection in image-to-diagram conversion.

## 5 EVALUATION METRICS

### 5.1 AUTOMATIC EVALUATION

We used four aspects of automatic evaluation: compilation success rate, image similarity, code similarity, and character similarity.

**Compilation success rate**  The compilation success rate (CSR) represents the percentage of generated Ti*k*Z codes that are successfully compiled into images. In this study, we employ two CSR metrics. The first is the averaged CSR, which calculates the ratio of successful compilations $N_{success}$ to the total number of generation attempts $N_{gen}$, expressed as $CSR_{avg} = \frac{N_{success}}{N_{gen}}$. This metric indicates how often a model succeeds in compilation on average. The second is the cumulative CSR, which represents the number of test samples that are compiled successfully through multiple iterations of iterative generation. It is defined as the ratio of successfully compiled samples, $N_{test\_success}$, to the total number of test samples, $N_{test}$, and is expressed as $CSR_{cum} = \frac{N_{test\_success}}{N_{test}}$. This metric shows the proportion of test samples that are correctly compiled through multiple attempts during iterative generation. Detailed examples are provided in Appendix J.

**Image similarity**  We used cosine similarity between image embeddings to measure the similarity between the generated image $I$ and the reference diagram image $I_r$. We used our D-SigLIP (see Sec. 4.3) for calculating image embeddings. We also calculated the image similarity score using the original CLIP model; however, the similarity score computed with CLIP correlated less with human evaluations than the similarity calculated using D-SigLIP. If the compilation failed, we set the similarity score to 0.

**Code similarity**  We used cosine similarity in the embedding space between $Y$ and $Y_r$. We generated the code embeddings using OpenAI's text embedding model.[6]

**Character similarity**  The character similarity calculates the similarity between the text in the generated image $I$ and the text in the reference image $I_r$ using Rouge-1 score (Lin, 2004). We used the OCR included in the Google Cloud Vision API to extract text. This metric indicates how well the model can read and generate text from the sketch.

---

[6]We used `text-embedding-3-small` version.

## 5.2 SUBJECTIVE EVALUATION

We conducted a subjective evaluation focusing on two key aspects: alignment and quality following established practices in previous studies (Otani et al., 2023; Ku et al., 2023). In our study, alignment measures the similarity between the generated and reference images, while quality assesses the coherence and appropriate arrangement of elements within the generated diagram. We employed a five-point scale for both metrics to ensure a nuanced evaluation.

**Alignment**   Annotators assessed alignment by visually comparing the generated diagram image $I$ to the reference diagram image $I_r$. The sketch diagram image $I_s$ was also provided for evaluation. Score of 1 and 5 indicated that the diagram's elements were completely misaligned and almost perfectly aligned, respectively. To illustrate a score of 1, a randomly selected rendered diagram image from the training dataset was displayed.

**Quality**   Annotators assessed the quality of the generated diagram images independently of the reference images, focusing on the structural clarity and arrangement of elements within the layout. A score of 1 was assigned to diagrams with poorly arranged, overlapping elements that were nearly unreadable. Conversely, a score of 5 was given to well-structured diagrams with logically arranged shapes and text that closely resembled human-created diagrams. The scale reflects the overall layout quality, ranging from incomprehensible to highly coherent visual representations.

**Annotation**   We comprehensively evaluated each model's outputs across the entire test set using Amazon Mechanical Turk. A total of 40 annotators conducted the annotation. For each test sample generated by each model, five annotators performed the evaluation. Diagrams that failed to be compiled were automatically assigned the minimum score of 1 for alignment and quality metrics. We computed the final score for each system and instance by averaging the three median evaluation scores, excluding potential outliers. A detailed description is provided in Appendix H.

## 6   EXPERIMENTAL SETUP

**Models for Comparison**   We evaluated several state-of-the-art models in our study.[7] GPT-4o, OpenAI's most efficient multimodal model. We also included GPT-4o mini, their top small model. From Anthropic, we employed Claude 3.5 Sonnet, the latest in their multimodal LLM series. Lastly, we assessed LLaVA-Next, a popular open-source model.

**Training parameters for IMGTIkZ**   We set the LoRA tuning parameters for training to $r$=128 and $\alpha$=256. Stage 1 training was conducted with a batch size of 256 for 6,000 steps. Stage 2 training used a batch size of 128 for 1 epoch. We used 8 A100 GPUs for training IMGTIkZ, and 1 H100 GPU for inference. More details are in Appendix B.

**Inference**   We applied iterative generation as the baseline for the four comparison models (see Sec.6), while for IMGTIkZ, we implemented both iterative and multi-candidate generation. The maximum number of attempts M for iterative sampling was set to 5, and the number of candidates K for multi-candidate generation was set to 20. More details are in Appendix A.

## 7   RESULTS

### 7.1   MAIN RESULTS

**Can models generate compilable TikZ code for diagrams?**   Table 3 presents the averaged CSR results (CSR_avg), with IMGTIkZ significantly outperforming the other models. The remaining models showed relatively low CSR_avg values (approximately 0.35-0.54), indicating insufficient adaptation to TikZ data. Since averaged CSR directly impacts user convenience, achieving higher scores is crucial. Figure 6 illustrates the progression of cumulative CSR across iterative generation attempts. IMGTIkZ achieved nearly 100% success after five attempts for the test data, while other methods leveled off at 0.8-0.9. These results indicate that 10-20% of samples remain uncompilable even after five attempts with these models.

---

[7]We used the `gpt-4o-2024-05-13` version for GPT-4o, the `gpt-4o-mini-2024-07-18` version for GPT-4o mini, the `claude-3-5-sonnet-20240620` version for Claude 3.5, and the `llama3-llava-next-8b` version, which is trained on the 8B Llama 3 model, for LLaVA-Next.

Table 3: The results of the automatic (0-1) and subjective (1-5) evaluations. The best results are highlighted in bold.

| Model | Automatic | | | | Subjective | |
|---|---|---|---|---|---|---|
| | ImageSim | CodeSim | CharSim | CSR_avg | Alignment | Quality |
| **Closed models** | | | | | | |
| GPT-4o | 0.695 | 0.821 | 0.611 | 0.479 | 3.00 | 3.20 |
| GPT-4o-mini | 0.595 | 0.814 | 0.514 | 0.376 | 2.39 | 2.71 |
| Claude 3.5 Sonnet | 0.753 | 0.813 | **0.671** | 0.544 | **3.32** | **3.54** |
| **Open-source models** | | | | | | |
| LLaVA-Next | 0.315 | 0.727 | 0.206 | 0.350 | 1.43 | 1.93 |
| IMGTI*k*Z-IG (*ours*) | 0.734 | 0.815 | 0.503 | 0.767 | 2.78 | 2.92 |
| IMGTI*k*Z-MCG (*ours*) | **0.821** | **0.822** | 0.594 | **0.799** | 3.13 | 3.30 |

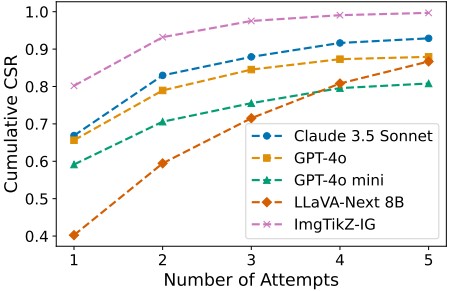

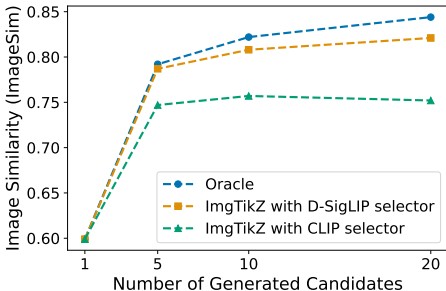

Figure 6: Progression of cumulative compilation success rate with varying number of attempts in an iterative generation.

Figure 7: Progression of image similarity with varying number of candidates in multi-candidate generation

**Can models generate diagram images close to the reference images?** ImageSim and Alignment in Table 3 present the similarity between generated and reference images. Claude 3.5 performed the best in Alignment score, followed by IMGTI*k*Z-MCG. In contrast, for ImageSim, IMGTI*k*Z-MCG outperformed the other models, with Claude 3.5 performing the second best. LLaVA-Next, which has a comparable model size to IMGTI*k*Z but lacks Ti*k*Z-specific training, performed poorly and rarely generated correct output. IMGTI*k*Z-MCG performed comparably to GPT-4o in Alignment despite being smaller, highlighting the effectiveness of our adaptation and multi-candidate generation strategy. However, even the best-performing model, Claude 3.5, achieved an average Alignment score of only 3.3, indicating that the generated diagrams match only 50-60% of the reference diagrams based on the subjective assessment. These results suggest that the task remains challenging, even for state-of-the-art models.

**Can models generate Ti*k*Z code close to the reference code?** Table 3 indicates that IMGTI*k*Z-MCG achieved the highest similarity scores for code similarity. However, code similarity scores are generally high with minimal inter-model differences. This indicates that high code similarity does not necessarily guarantee quality image generation. This discrepancy highlights a critical insight for model training: generating code that closely resembles the ground truth is insufficient. Similar to conventional VLMs, IMGTI*k*Z training relies on loss based on the next-word prediction of code. However, our findings suggest image similarity metrics need to be incorporated in training or inference phrases. This result aligns with the significant performance improvements of IMGTI*k*Z-MCG.

**Can models accurately render text in sketch images?** The CharSim in Table 3 provides insight into each model's ability to recognize characters in sketch images and render them accurately in Ti*k*Z diagram. Claude 3.5 achieved the highest CharSim score, followed by GPT-4o. While IMGTI*k*Z performed comparably to GPT-4o in Alignment, it significantly underperforms in CharSim. This suggests that IMGTI*k*Z has enhanced diagram shape recognition but struggles with detailed character recognition. This limitation may reflect the resolution constraints of the SigLIP vision encoder. However, the substantial improvement in CharSim with multi-candidate generation indicates character recognition needs to be strengthened during training.

Table 4: Evaluation of the effectiveness of SKETIkZ as training data.

| Model | ImageSim | CharSim | CSR_avg |
|---|---|---|---|
| IMGTIkZ-IG | 0.734 | 0.502 | 0.767 |
| w/ SKETIkZ only | 0.513 | 0.358 | 0.533 |
| LLaVA-Next 8B | 0.315 | 0.205 | 0.350 |

Table 5: Effectiveness of two data augmentation: (a) ImgAugTikZ and (b) AugTikZ.

| Model | ImageSim | CharSim | CSR_avg |
|---|---|---|---|
| IMGTIkZ-IG | 0.734 | 0.502 | 0.767 |
| w/o (a) | 0.668 | 0.457 | 0.635 |
| w/o (a) and (b) | 0.601 | 0.439 | 0.541 |

**Can models generate high-quality diagrams?** Table 3 presents quality scores from subjective evaluations. Claude 3.5 achieved the highest average score of 3.54 out of 5, followed by IMGTIkZ-MCG. Even the best-performing Claude model produces approximately 38% of samples with quality scores below 3 (indicating significant overlap of shapes and text), demonstrating that current VLMs still struggle with correct diagram layout rendering. This limitation in spatial reasoning is a common challenge among current VLMs. Our task and dataset can be considered one of the benchmark datasets for evaluating VLMs' spatial reasoning capabilities.

**How does the number of candidates in multi-candidate generation affect performance?** Figure 7 illustrates the image similarity trends for ImgTikZ-MCG as the number of candidates $K$ in multi-candidate generation varies. The oracle represents the highest achievable performance by selecting the best candidate on the basis of image similarity to the reference diagram $I_r$. Results show performance significantly improved when candidates were increased from one to five. Both oracle and IMGTIkZ demonstrate enhanced image similarity with more candidates. However, when replacing the selection model from D-SigLIP to CLIP, performance does not increase beyond five candidates. This indicates the importance of selection model quality in multi-candidate generation.

**Do subjective evaluations correlate with automated evaluations?** We analyzed correlations between subjective alignment ratings and automatic evaluation metrics. Pearson's correlation coefficients were calculated between human-rated alignment and image similarity (0.759), code similarity (0.365), and character similarity (0.592). Image similarity correlated strongly with the subjective evaluation, while code similarity correlated weakly with it. Character similarity correlated moderately, highlighting the importance of textual information in diagram evaluation. Image similarity metrics often fail to capture this local textual similarity.

**Are the subjective evaluations consistent?** To assess inter-annotator agreement in subjective evaluations, we employed Krippendorff's $\alpha$ (Krippendorff, 1980), a measure commonly used in related research (Otani et al., 2023; Ku et al., 2023). The analysis showed Krippendorff's $\alpha$ of 0.761 for alignment and 0.662 for quality, indicating substantial to moderate agreement among annotators in their subjective assessments.

## 7.2 DETAILED ANALYSIS

**How effective is SKETIkZ alone as training data?** We evaluated the effectiveness of our SKETIkZ dataset, comprising only 2.6k hand-drawn sketch samples, as training data. We evaluated the performance of a model trained solely on SKETIkZ in step 2. Results are presented in Table 4. While the SKETIkZ-only model underperforms compared to the full-data model, it significantly outperforms LLaVA-Next, indicating meaningful adaptation even with this limited dataset.

**Is data augmentation effective?** To assess the impact of our two data augmentation methods, we trained models excluding ImgAugTikZ and both ImgAugTikZ and AugTikZ. Results are presented in Table 5. The observed significant decrease in image similarity, character similarity, and CSR_avg when excluding these datasets demonstrates the effectiveness of both augmentation methods.

**To what extent does image augmentation improve sketch recognition?** While the ablation study in Table 5 confirmed image augmentation improved performance, we further investigated its impact on sketch recognition. Specifically, we compared the performance gap between using rendered reference images $I_r$ and sketch images $I_s$ as input. The closer the performance of sketch input approaches that of rendered image input, the more robust the model's understanding of sketch noise can be considered. Results are shown in Table 6. Without ImgAugTikZ, image similarity decreased by approximately 12.5% and character similarity by 22.7%. In contrast, ImgTikZ limited these reductions to 6.97% for image similarity and 17.0% for character similarity. However, ImgTikZ still

Table 6: Performance gap between rendered and sketch image inputs: comparison IMGTi*k*Z-IG and IMGTi*k*Z-IG without ImgAugTi*k*Z data.

| Metric | Model | |
|---|---|---|
| | **IMGTi*k*Z-IG** | **w/o ImgAugTi*k*Z** |
| **ImageSim** | | |
| Rendered Image | **0.789** | 0.763 |
| Sketch Image | **0.734** | 0.668 |
| Performance Gap | **-6.97%** | -12.5% |
| **CharSim** | | |
| Rendered Image | **0.605** | 0.591 |
| Sketch Image | **0.503** | 0.457 |
| Performance Gap | **-16.9%** | -22.7% |

Table 7: Performance gap between rendered and sketch image inputs across different sketching tools. Evaluation conducted using the IMGTi*k*Z-IG.

| Metric | Tool | | |
|---|---|---|---|
| | **Paper** | **Whiteboard** | **Tablet** |
| **ImageSim** | | | |
| Rendered Image | 0.793 | 0.796 | 0.754 |
| Sketch Image | 0.735 | 0.716 | 0.740 |
| Performance Gap | **-7.31%** | **-10.1%** | **-1.90%** |
| **CharSim** | | | |
| Rendered Image | 0.587 | 0.627 | 0.581 |
| Sketch Image | 0.502 | 0.451 | 0.570 |
| Performance Gap | **-14.5%** | **-28.1%** | **-1.89%** |

does not match rendered image input performance, suggesting the potential for further improving performance by constructing a more noise-robust model construction.

**Does image augmentation improve performance for non-sketch images?** Comparing ImageSim and CharSim results for Rendered Images in Table 6 reveals that ImgTi*k*Z outperforms the model without image augmentation. Image augmentation enhanced both ImageSim (0.763→0.789) and CharSim (0.591→0.605) scores, showing improved recognition even for clean, computer-rendered images.

**Does image recognition difficulty vary across sketch tools?** Table 7 presents the performance gap in image and character similarity when using rendered images versus sketches as inputs across different sketching tools. Results show that tablet sketches maintain image and character similarity close to rendered images. However, sketches from paper and whiteboard tools show significant performance degradation, declining by 7-10% in image similarity and 14-28% in character similarity. This performance drop suggests that paper and whiteboard sketches are more challenging for the model to process, likely due to their greater noise variety than tablet sketches. Whiteboard sketches showed the most significantly in performance. While our image augmentation techniques have relatively minimized the gap with rendered image input, further performance improvements will require developing methods more robust to real-world noise.

## 8 CONCLUSION

We introduced SKETi*k*Z, a benchmark dataset with 3,231 pairs of hand-drawn sketches and their corresponding Ti*k*Z codes for generating diagrams. Our experiments demonstrate that current VLMs face considerable challenges in this task, highlighting the value of SKETi*k*Z as a benchmark for future research. We also developed IMGTi*k*Z, an image-to-TikZ model. Despite being smaller, this model performed as well as GPT-4o in subjective evaluations. This success came from using two data augmentation techniques and generating multiple candidates during inference. SKETi*k*Z is publicly available, and we expect these data resources and insights to drive the development of more advanced and efficient methods for automating vector graphics creation from hand-drawn sketches.

## 9 LIMITATION

Currently, SKETi*k*Z is restricted to generating diagrams using Ti*k*Z. However, the methodology could be extended to other formats such as SVG, HTML, Python, and JavaScript for diagram generation from code. Exploring these additional formats could enhance the dataset's generality and applicability. Transforming sketches into well-formed diagrams involves information completion, which can potentially lead to hallucination. An important direction for future work is developing an interactive system that allows users to modify generated diagrams through instructions. Furthermore, while our multi-candidate generation strategy considers code correctness and image quality metrics after code generation, incorporating these metrics directly into the training phase could potentially lead to better generation results, representing a promising direction for future work.

## ETHICS STATEMENT

**Were annotators for sketch creation told what the dataset would be used for, and did they consent?** Yes. BAOBAB Inc. was fully responsible for managing the annotators. BAOBAB Inc. provides task descriptions, training, and agreements for each project with the annotators `https://baobab-trees.com/en/service`.

**Data License** SKETIkZ is derived from a publicly available subset of DaTikZ (Belouadi et al., 2023), which permits copying and redistributing content under a Creative Commons Attribution License,[8] the GNU Free Documentation License,[9] or the MIT License.[10]

**Potential ethical considerations** We believe that there are minimal ethical considerations within the scope of this current research. However, as more accurate automatic diagram generation becomes feasible in the future, several issues may arise. These potential problems include the misuse of highly accurate auto-generated diagrams to spread misinformation, the risk of AI models perpetuating or amplifying biases from their training data, and the possibility of advanced systems inadvertently reproducing copyrighted diagram designs, thereby raising intellectual property and copyright infringement issues; all of these challenges necessitate the establishment of appropriate guidelines to address them effectively.

## REPRODUCIBILITY STATEMENT

**Dataset Distribution** SKETIkZ is available at `https://sketikz.github.io/`

**Details of models, hyperparameters, and manual evaluation** Appendices B, C, and E provide detailed information about the models developed in this study. Appendix A describes the specifics of our inference process. Appendix H presents details regarding the subjective evaluation. Additionally, Appendices D and G presents details of the data creation process.

## ACKNOWLEDGMENTS

This work was supported by JSPS KAKENHI Grant Numbers, 21K21343 and 24K20829, and JST Moonshot R&D Program Grant Number JPMJMS2236. We thank BAOBAB Inc. for creating hand-drawn diagram images with high quality and accuracy. In this research work, we used the "mdx: a platform for building data-empowered society". This study was carried out using the TSUBAME4.0 supercomputer at Institute of Science Tokyo.

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

## A  DETAILS OF INFERENCE

**Inference Procedure**   We used pdflatex from TeX Live 2023[11] to compile generated TikZ code into a diagram image. We first cropped the rendered image using pdfcrop and then converted it to a PNG file to calculate image similarity.

**Hyperparameters for closed models**   We used the API's default parameters for the closed models GPT-4o, GPT-4o mini, and Claude. The $max\_token$ parameter was set to 2,048 for all models.

**Hyperparameters for LLaVA1.6 and IMGTikZ**   We set the maximum number of newly generated tokens to 2,048 and generated the code through sampling. The sampling temperature was set to 0.6, a value determined through evaluation using the validation set.

## B  HYPERPARAMETERS FOR TRAINING IMGTikZ

We conducted the training using the official code of LLaVA.[14] Table 8 details the hyperparameters used for stage 2 training of IMGTikZ. For stage 2 training, we used a total batch size of 128. The stage 1 training employed similar hyperparameters, with a few exceptions: we set the batch size to 32 with gradient accumulation over 4 steps, resulting in a total batch size of 128, and we increased the max_length to 2048. These parameters were derived from the original implementation of LLaVA1.5. The training process consisted of 6000 steps for stage 1 and a full epoch for stage 2. We conducted the training using 8 A100 GPUs. The total training time was approximately 24 hours for stage 1 and 60 hours for stage 2.

---

[11]`https://tug.org/texlive/`
[14]https://github.com/haotian-liu/LLaVA

Table 8: Configuration for the IMGTI*k*Z model training.

| Option | Value |
|---|---|
| model_name (LLM) | `deepseek-ai/deepseek-coder-6.7b-instruct`[12] |
| model_name (Vision encoder) | `google/siglip-so400m-patch14-384`[13] |
| lora_r | 128 |
| lora_alpha | 256 |
| mm_projector_lr | 2e-5 |
| mm_projector_type | mlp2x_gelu |
| group_by_modality_length | True |
| bf16 | True |
| num_train_epochs | 1 |
| batch_size | 16 |
| gradient_accumulation_steps | 8 |
| weight_decay | 0 |
| warmup_ratio | 0.03 |
| lr_scheduler_type | cosine |
| model_max_length | 4096 |
| gradient_checkpointing | True |

## C  D-SIGLIP: AN SIGLIP MODEL ADAPTED FOR DIAGRAM

We trained D-SigLIP using a contrastive learning framework based on Hugging Face's code.[15] We used the `google/siglip-so400m-patch14-384` version of SigLIP as the vision encoder. During training, we applied augmentation twice to each image, aiming to maximize the similarity between augmented versions of the same image within the batch. Image augmentation was performed on-the-fly using imgaug.[16] The noise pipeline applied through imgaug is detailed below.

Listing 1: Image Augmentation Pipeline for D-SigLIP Training

```
pipeline = iaa.Sequential([
    iaa.Affine(scale={"x": (0.7, 1.0), "y": (0.7, 1.0)}, cval=255),
    iaa.Affine(rotate=(-5, 5), cval=255),
    iaa.Affine(translate_percent={"x": (-0.1, 0.1), "y": (-0.1, 0.1)},
        cval=255),
    iaa.Sometimes(0.2, iaa.ChangeColorTemperature((1100, 3000))),
    iaa.Sometimes(0.3, iaa.AdditiveGaussianNoise(scale=(10, 20))),
    iaa.Sometimes(0.3, iaa.MultiplyAndAddToBrightness(mul=(0.8, 1.2), add
        =(-5, 5))),
    iaa.Sometimes(0.3, iaa.GammaContrast((0.8, 1.2))),
    iaa.Sometimes(0.3,
        iaa.BlendAlphaSimplexNoise(
        iaa.Multiply((1.5, 2.5), per_channel=True),
        upscale_method='cubic',
        iterations=(1, 2)
    )),
    iaa.Sometimes(0.1, iaa.LinearContrast((0.8, 1.2))),
    iaa.ElasticTransformation(alpha=(15.0, 40.0), sigma=(5.0, 10.0)),
])
```

The training was conducted using four H100 80G GPUs. We set the batch size to 1024, the learning rate to 5e-5, and the warmup steps to 0, with training carried out for 200 steps.

## D  DATASET COLLECTION PROCESS

First, we compiled the TikZ code from DaTi*k*Z (Belouadi et al., 2023) to render the diagram images. Then, we developed a diagram classification model (See Section E) using the ACL-fig (Karishma

---

[15]https://github.com/huggingface/transformers/tree/main/examples/pytorch/contrastive-image-text

[16]https://imgaug.readthedocs.io/en/latest/

et al., 2023) data, which was subsequently employed to classify the rendered diagrams from the DaTi*k*Z dataset. We then extracted diagrams with the predicted labels Tree, Graph, Architecture diagram, Neural networks, and Venn diagram and sampled 4,000 instances from them.

BAOBAB Inc. coordinated multiple annotators to create the corresponding sketches for sampled instances. We excluded diagrams that were too complex to be sketched, diagrams of bar charts and line graphs that require numerical data, overly simplistic diagrams comprising only straight lines or dots, diagrams with illegible text, diagrams containing non-English text, and incomplete diagrams that were unnaturally truncated from the tasks during this process. The annotators selected one of the following tools to create the sketches: paper, whiteboard, or tablet. When using paper or whiteboard, they captured photos of the hand-drawn images with a smartphone camera. They used the drawing tool's save function for tablets to save the images. All images were then converted to PNG format. As a result of these processes, we ultimately created 3,231 instances.

## E    DIAGRAM IMAGE CLASSIFICATION MODEL FOR DATA CONSTRUCTION

We developed a model to classify diagram images into categories by fine-tuning a pre-trained vision transformer on the ACL-fig dataset.[17]   For the pre-trained VIT, we used Google's `vit-large-patch16-224-in21k`.[18]   The training was conducted using Hugging Face's tools.[19]   The parameters used for the training are listed in Table 9. We trained the model using a NVIDIA A100 GPU. The model achieved a classification accuracy of 0.886 on the evaluation dataset.

Table 9: Configuration for the image classification model.

| Option | Value |
|---|---|
| model_name | `google/vit-large-patch16-224-in21k` |
| learning_rate | 2e-5 |
| num_train_epochs | 5 |
| batch_size | 8 |
| warmup_ratio | 0 |
| weight_decay | 0 |

Table 10 presents the breakdown of estimated image labels within the sampled data. Furthermore, Figure 8 illustrates example diagrams for each estimated label category. While these are estimated labels and may potentially include diagrams that do not strictly conform to any specific category or contain estimation errors, we confirmed that there are diverse types of diagrams in our dataset.

## F    SKETCH IMAGE EXAMPLES

Figure 9 shows a subset of the collected sketch images.

## G    DETAILS OF THE DATA AUGMENTATION

### G.1    AUGTI*k*Z: THE AUGMENTATION FOR INCREASING DIAGRAM VARIATION

From the arXiv source files,[20] we initially obtained 916,123 TikZ code samples. However, only 155K of these were successfully compiled. We utilized these compilable codes as RenderTi*k*Z. While the remaining codes failed to compile, we recognized their potential to significantly increase diagram variations if effectively utilized. To achieve this, we employed two types of augmentation prompts. The first prompt focused on code revision and was applied to the initially failed compilations. The second prompt, aimed at code modification, was applied to the entire dataset. The specific instructions provided were as follows. We used the `gpt-3.5-turbo-0125` version of GPT-3.5 to create the augmentation data.

---

[17]`https://huggingface.co/datasets/citeseerx/ACL-fig`

[18]`https://huggingface.co/google/vit-large-patch16-224-in21k`

[19]`https://github.com/huggingface/transformers/blob/main/examples/pytorch/image-classification/run_image_classification.py`

[20]`https://info.arxiv.org/help/bulk_data_s3.html`

Table 10: Proportion of estimated image labels in the sampled data.

| Category | Number | Proportion |
|---|---|---|
| Tree | 1,799 | 45.0 % |
| Graph | 1,046 | 26.2 % |
| Architecture diagram | 646 | 16.2 % |
| Neural networks | 459 | 11.5 % |
| Venn diagram | 50 | 1.1 % |
| All | 4,000 | 100% |

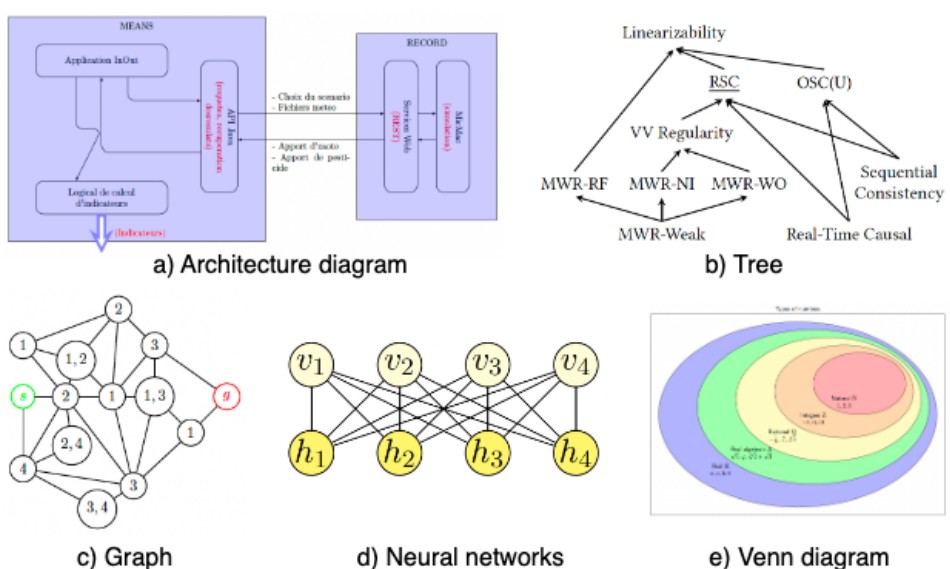

Figure 8: Examples of estimated image labels and their diagrams.

> **Prompts for data augmentation**
>
> - Please modify the given LaTeX source code to make it compilable, including only the required preamble statements. If any external files are referenced, please modify the code to avoid referencing external files and include the content directly. The output should consist solely of the code itself, without any supplementary text.
> - Please generate Ti*k*Z source code that modifies parts of the following code to create a different diagram. Ensure the code is compilable and includes only the required preamble statements. If any external files are referenced, please modify the code to avoid referencing external files and include the content directly. The output should consist solely of the code itself, without any supplementary text.

We included only the code that successfully compiled and rendered images correctly in our dataset AugTi*k*Z. Furthermore, we excluded images that were rendered at extreme scales (either too large or too small) from the training dataset.

### G.2    IMGAUGTI*k*Z: THE AUGMENTATION FOR INCREASING IMAGE VARIATION

To simulate the noise typically present in sketches, we applied several augmentation techniques to both RenderTi*k*Z and AugTi*k*Z. These included compositing with notebook background images, augmentation using imgaug, and white balance augmentation.[21]  For notebook backgrounds, we created eight unique images independently of the sketch annotation process. The imgaug library

---

[21]https://github.com/mahmoudnafifi/WB_color_augmenter

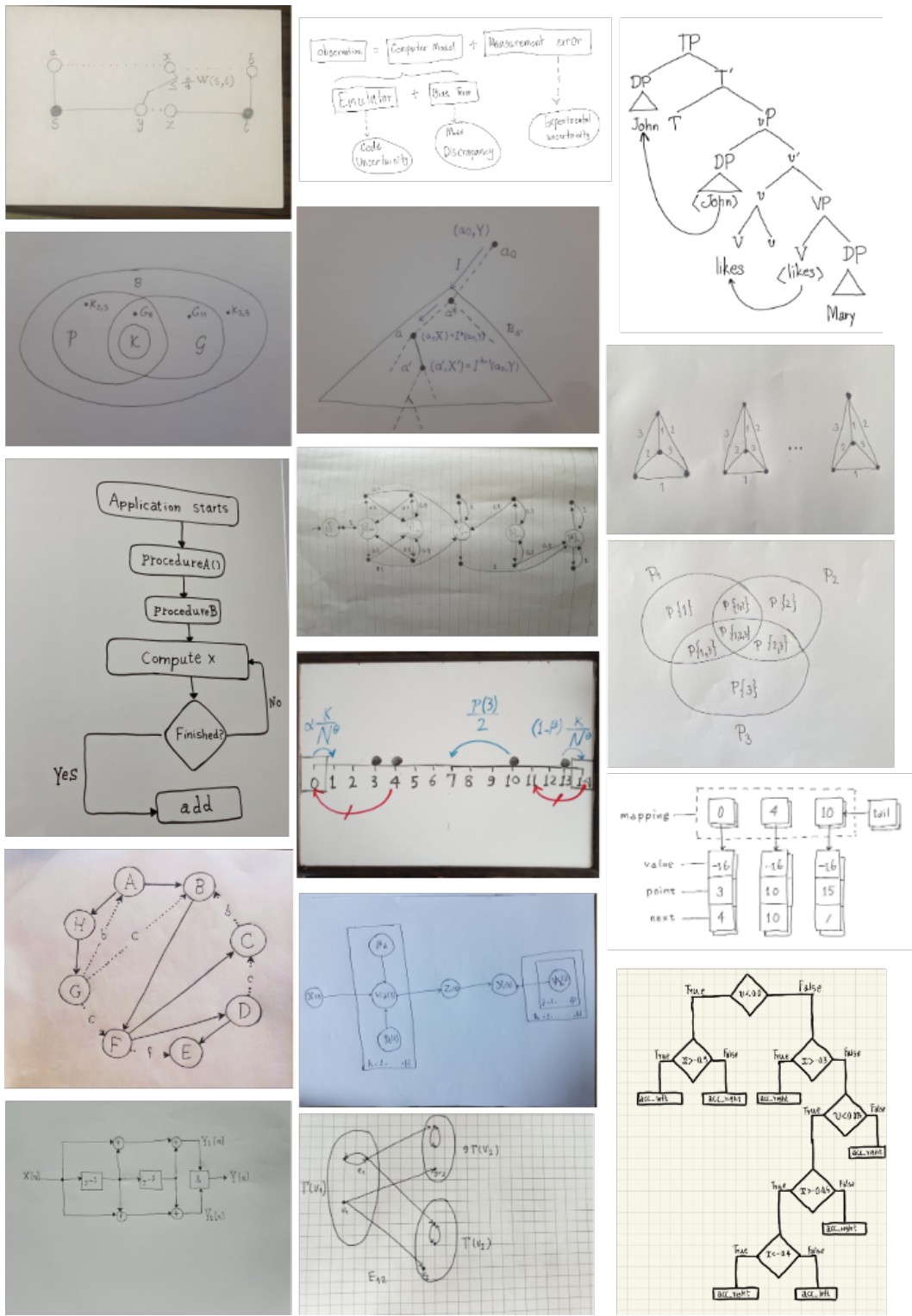

Figure 9: Examples of collected sketch images.

was used to generate variations in rotation, distortion, Gaussian noise, brightness, and contrast. The specific augmentation pipeline created with imgaug is detailed below.

Listing 2: Image Augmentation Pipeline for Image Augmentation

```
pipeline = iaa.Sequential([
    iaa.Pad(percent=0.3, pad_mode="median"),
    iaa.Sometimes(0.3, iaa.AdditiveGaussianNoise(scale=(10, 20))),
    iaa.Sometimes(0.3, iaa.MultiplyAndAddToBrightness(mul=(0.8, 1.2), add
        =(-5, 5))),
    iaa.Sometimes(0.3, iaa.GammaContrast((0.8, 1.2))),
    iaa.Sometimes(0.3,
        iaa.BlendAlphaSimplexNoise(
        iaa.Multiply((1.5, 2.5), per_channel=True),
        upscale_method='cubic',
        iterations=(1, 2)
    )),
    iaa.Sometimes(0.1, iaa.LinearContrast((0.8, 1.2))),
    iaa.Affine(rotate=(-5, 5)),
    iaa.ElasticTransformation(alpha=(15.0, 30.0), sigma=(5.0, 10.0)),
    iaa.CropToFixedSize(width=int(width*0.8), height=int(height*0.8))
])
```

## H  SUBJECTIVE EVALUATION

For each test sample, annotators evaluated the alignment and quality of the six systems' outputs, GPT-4o, GPT-4o mini, Claude 3.5 Sonnet, LLaVA-Next, IMGTIkZ-IG, IMGTIkZ-MCG. We compensated annotators at a rate of \$1.5 per test sample.
We provided annotators with the following instructions for conducting their evaluations:

> **Instructions**
>
> For each image A-F, please assign a score from 1 to 5 based on the following two aspects. You may also use 0.5 increments, such as 1.5 or 3.5.
>
> - **Alignment:** The extent to which the generated diagram image matches the layout and content of the hand-drawn image.
>
> - **Quality:** The overall completeness of the generated diagram image, regardless of the presence or absence of the hand-drawn and reference image.

The specific evaluation criteria for alignment that we instructed the annotators to follow are as follows:

> **Evaluation Criteria for Alignment**
>
> **1:** The elements of the diagram in the generated image and the hand-drawn image do not match at all.
>
> **2:** The elements of the diagram in the generated image and the hand-drawn image match approximately 25%.
>
> **3:** The elements of the diagram in the generated image and the hand-drawn image match approximately 50%.
>
> **4:** The elements of the diagram in the generated image and the hand-drawn image match approximately 75%.
>
> **5:** The elements of the diagram in the generated image and the hand-drawn image match almost perfectly.

The specific evaluation criteria for quality that we instructed the annotators to follow are as follows:

> **Evaluation Criteria for Quality**
>
> **1:** Almost complete overlap of text or shapes, making the diagram unreadable.
>
> **2:** Significant overlap of text or shapes, and the arrangement of elements is unnatural.
>
> **3:** Significant overlap of text or shapes, making some elements unreadable, or some elements are arranged unnaturally.
>
> **4:** Some overlap of text or shapes, but the arrangement of elements is neat.
>
> **5:** No overlap of text or shapes, and the arrangement of elements is as neat as a human-created diagram.

Figure 10 presents a partial screenshot of the annotation system interface. The complete template file for the annotation system, which includes all instructions, can be accessed this link `https://storage.googleapis.com/sketikz/template_202409_example.html`.

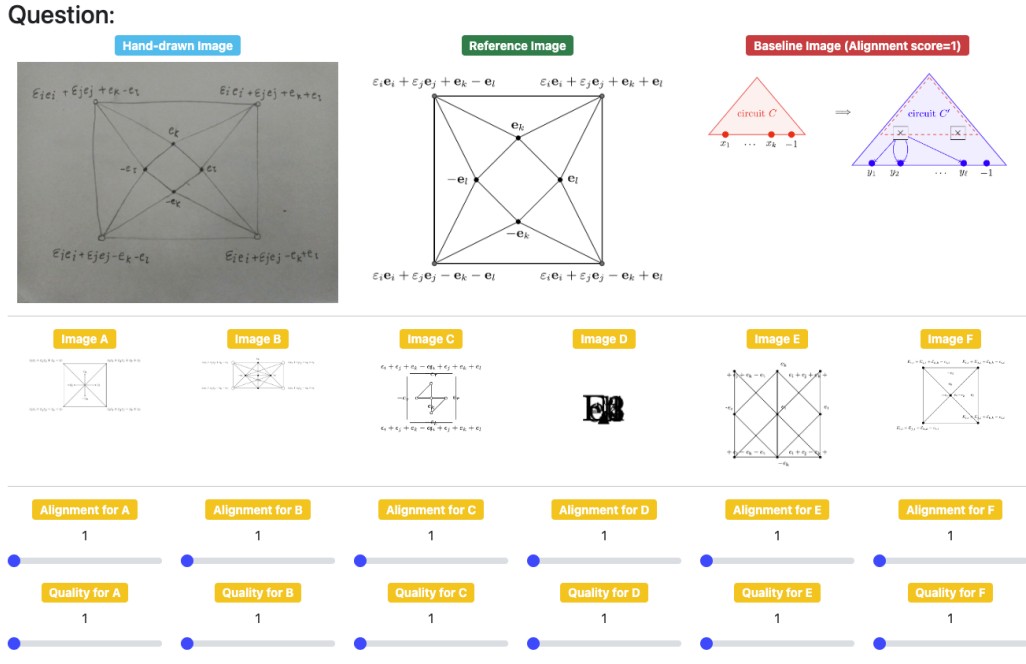

Figure 10: Screenshot of the annotation interface: In the HTML, each image can be clicked to enlarge, allowing annotators to view the details of the diagrams.

## I   GENERATED DIAGRAM EXAMPLES WITH EVALUATION SCORES

Tables 11 and 12 show some examples of generated diagrams. IMGTI$k$Z-MCG generally selects better candidates compared to IMGTI$k$Z-IG.

## J   DETAILED EXPLANATION OF COMPILATION SUCCESS RATE (CSR)

To better illustrate the difference between $CSR_{\text{avg}}$ and $CSR_{\text{cum}}$, we provide examples below. $CSR_{\text{avg}}$ represents the success rate across all generation attempts. For example, if a model attempts $N$ generations for each of the 100 test samples and succeeds in compilation $K$ times, then

$$CSR_{\text{avg}} = \frac{N_{\text{success}}}{N_{\text{gen}}} = \frac{K}{(100 \times N)}. \tag{1}$$

Table 11: Examples of generated diagrams and their metric scores.⊠ indicates a compile error and, therefore, has no score.

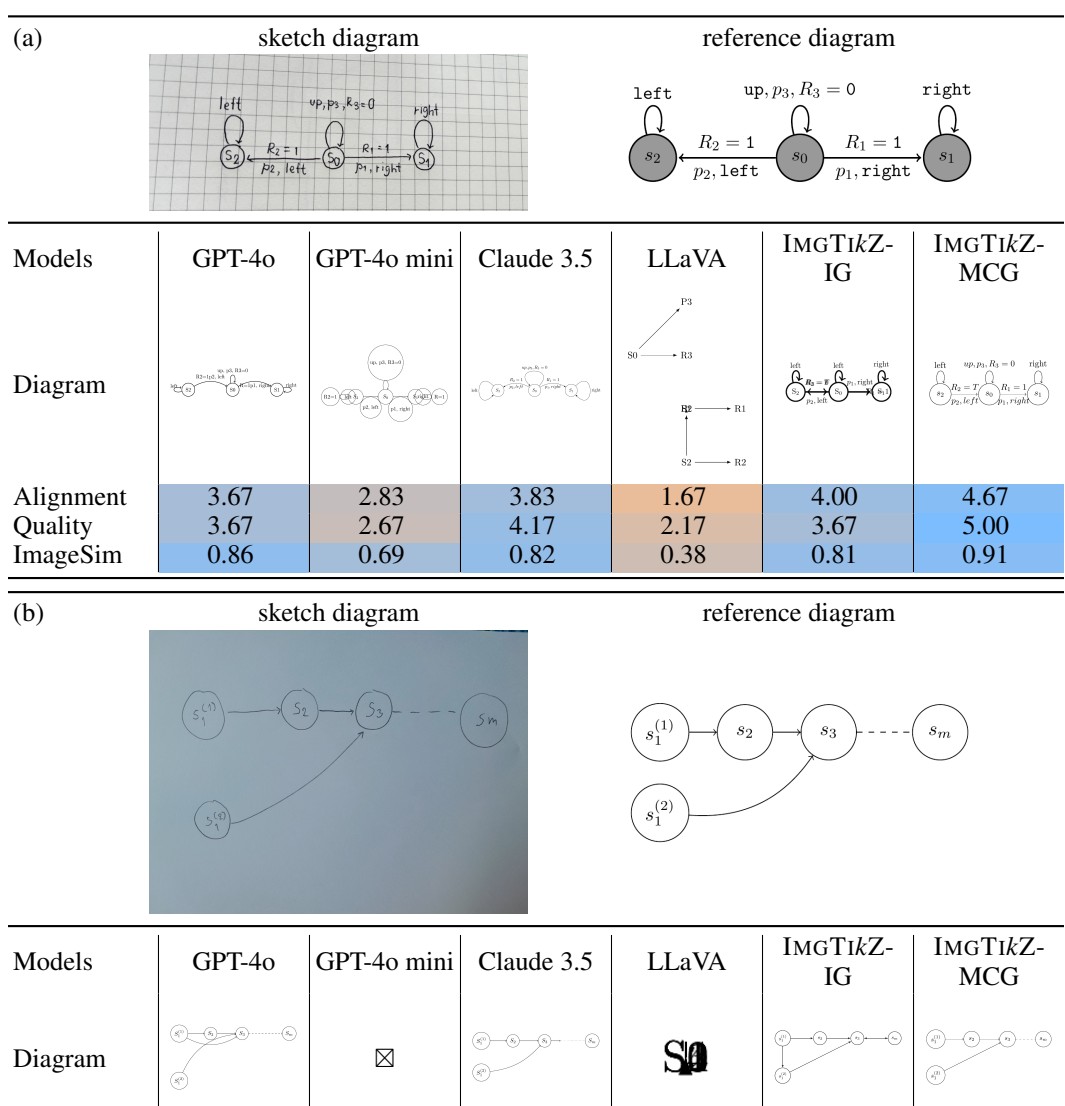

| Models | GPT-4o | GPT-4o mini | Claude 3.5 | LLaVA | IMGTIkZ-IG | IMGTIkZ-MCG |
|---|---|---|---|---|---|---|
| Alignment | 3.67 | 2.83 | 3.83 | 1.67 | 4.00 | 4.67 |
| Quality | 3.67 | 2.67 | 4.17 | 2.17 | 3.67 | 5.00 |
| ImageSim | 0.86 | 0.69 | 0.82 | 0.38 | 0.81 | 0.91 |

| Models | GPT-4o | GPT-4o mini | Claude 3.5 | LLaVA | IMGTIkZ-IG | IMGTIkZ-MCG |
|---|---|---|---|---|---|---|
| Alignment | 3.83 | N/A | 4.50 | 1.00 | 4.17 | 4.67 |
| Quality | 4.00 | N/A | 4.83 | 1.00 | 4.83 | 4.83 |
| ImageSim | 0.79 | N/A | 0.92 | 0.05 | 0.87 | 0.92 |

To illustrate, if we make 10 generation attempts for each of the 100 test samples (totaling 1,000 generations) and achieve successful compilation in 400 cases, then

$$CSR_{\text{avg}} = \frac{400}{1000} = 0.4. \tag{2}$$

$CSR_{\text{cum}}$, which is exclusively used for iterative generation, measures the cumulative proportion of test samples achieving successful compilation across multiple attempts. Consider the following sequential process for 100 test samples:

- First generation: 50 of the 100 samples compile successfully
- Second generation: 20 of the remaining 50 (100 - 50) samples compile successfully
- Third generation: 10 of the remaining 30 (50 - 20) samples compile successfully

Table 12: Examples of generated diagrams and their metric scores.⊠ indicates a compile error and, therefore, has no score.

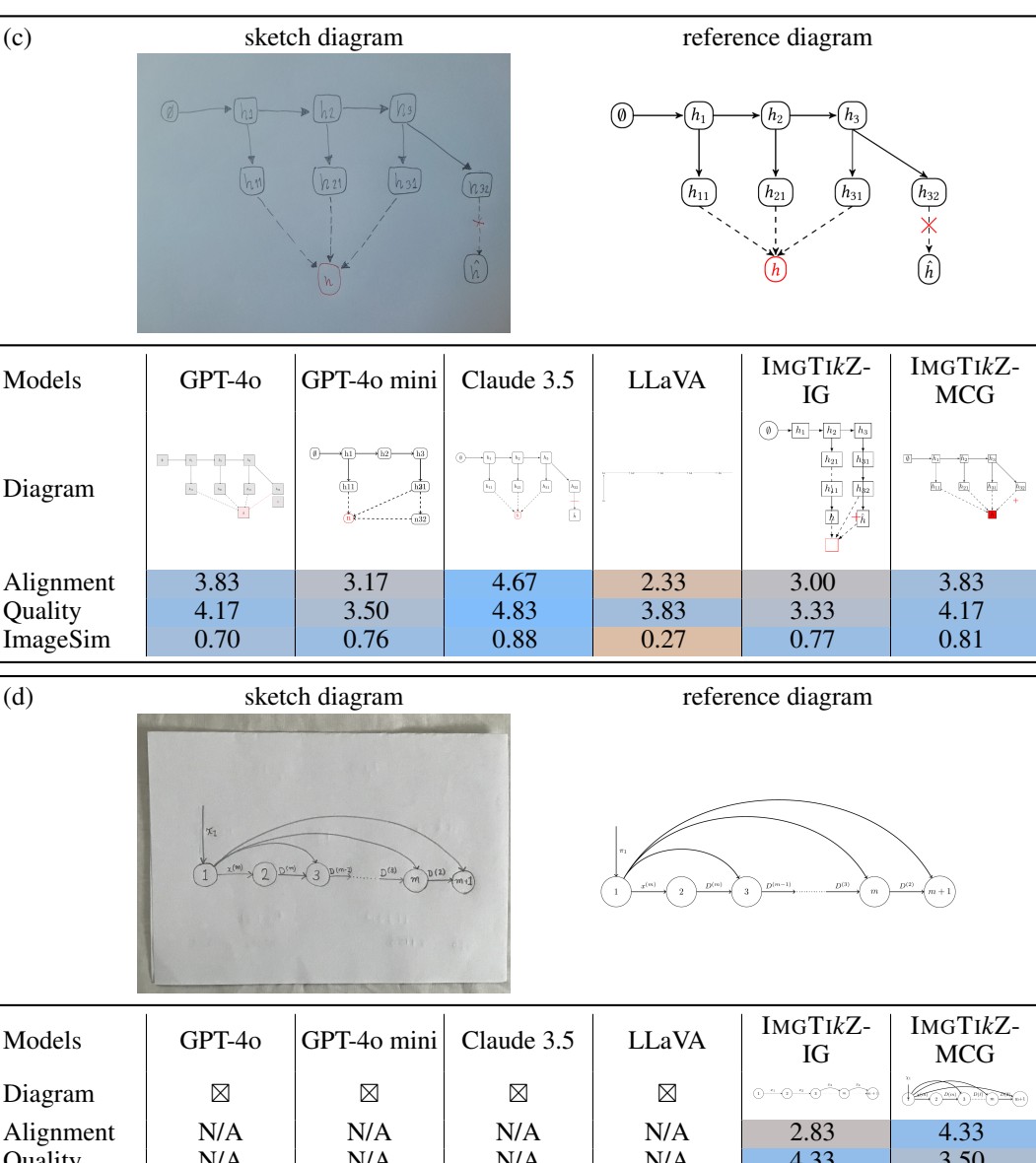

In this scenario,

$$CSR_{\text{cum}} = \frac{N_{\text{test\_success}}}{N_{\text{test}}} = \frac{50 + 20 + 10}{100} = 0.8. \qquad (3)$$

This metric specifically quantifies the proportion of test samples that eventually achieve successful compilation, independent of the total generation attempts.

The motivation for utilizing these two distinct evaluation metrics arises from their complementary analytical perspectives: $CSR_{\text{avg}}$ represents the average compilation success rate, enabling fair model comparison. $CSR_{\text{cum}}$ measures the proportion of successfully compiled test samples across multiple attempts, analogous to a recall metric.

Table 13: Comparison of IG and Step-by-step + IG approaches.

| Model | IG | Two-stage + IG |
|---|---|---|
| GPT-4o | 0.695 | 0.730 |
| Claude 3.5 Sonnet | 0.753 | 0.733 |

## K  EVALUATION OF THE EFFECTIVENESS OF A STEP-BY-STEP APPROACH: TEXT GENERATION FOLLOWED BY CODE GENERATION

Although this paper does not focus on this approach, we also investigate a step-by-step method for solving this task, where a textual description of the sketch image is first generated, followed by code generation. To assess the effectiveness of this approach, we conducted experiments using Claude 3.5 Sonnet and GPT-4o. The results are presented in Table 13. The evaluation was performed using ImageSim for automated evaluation. The results indicate that while Claude 3.5 did not improve performance, GPT-4o slightly increased performance. This suggests that text-mediated generation is beneficial for models with relatively lower code generation capabilities but has limited impact on models with stronger code generation abilities. One possible reason is that errors may occur during text generation, meaning that textual descriptions do not always positively contribute to subsequent code generation. Investigating more effective step-by-step methods for models with sufficiently high code generation capabilities remains an important direction for future research.

