# OpenReview forum: "Sketch2Diagram: Generating Vector Diagrams from Hand-Drawn Sketches"
_ICLR.cc/2025/Conference — ICLR 2025 Poster_

### Official Review · Reviewer_SFio · 2024-10-31

**Soundness:** 3
**Presentation:** 3
**Contribution:** 3
**Rating:** 6
**Confidence:** 4

**Summary:**

This article provides a solution for the task of generating high-quality vector graphics from hand-drawn sketches, a convenient method for conveying complex concepts across various fields. However, the problem of converting hand-drawn sketches to vector graphics remains inadequately addressed due to the lack of datasets. To address this issue, the article constructs a dataset called SKETIkZ, which contains 3,231 pairs of hand-drawn sketches, reference images, and corresponding TikZ code. The authors also commit to making the SKETIkZ dataset fully publicly available. Leveraging the SKETIkZ dataset, the article proposes a modest-sized method called IMGTIkZ that can compete with the performance of GPT-4o.

**Strengths:**

1. This article is written in a clear and concise manner, introducing the proposed method very clearly and discussing in detail some issues of concern to readers in the results section.
2. The article has open-sourced a dataset for converting hand-drawn sketches to vector graphics, contributing usable foundational data for subsequent research on the same task.

**Weaknesses:**

1. The method proposed in this article for converting hand-drawn sketches to vector graphics relies more on the combination of existing technical solutions and does not introduce particularly innovative technical approaches.
2. This article carries out extensive data augmentation and supplementation when using the SKETIkZ dataset, which raises curiosity about whether the SKETIkZ dataset itself could play a significant role in future research endeavors.

**Questions:**

1. I find the SKETIkZ dataset contributed by this article to be quite beneficial, which is why I am more concerned about this dataset. Although I have noticed that the article discusses how the use of the SKETIkZ dataset alone can enhance model performance, I am somewhat concerned about whether the SKETIkZ dataset will universally bring performance improvements in future research work.

2. Regarding the vector graphics generated from hand-drawn sketches, if there are minor errors in the output, I wonder if there are any simple methods available for correction.

---

> ### Author Response · Authors · 2024-11-19
>
> We appreciate your positive comments regarding the value of the dataset and clarity of writing. The following responses address the questions and concerns raised.
>
> > ### Whether the SKETIkZ dataset itself could play a significant role in future research endeavors.
>
> Based on your insightful suggestions, we will expand our manuscript to address the points below, offering a more detailed examination of SkeTikZ's value and potential impact.
>
> ### 1. Value as an Evaluation Benchmark Dataset
> The SkeTikZ dataset serves not only as training data but also as a valuable evaluation benchmark. It is particularly useful because there are very few datasets that pair human sketches with their corresponding source code and images. SkeTikZ includes sketches from various tools (paper, whiteboards, tablets), allowing us to evaluate across diverse input environments.
>
> While using synthetic data has greatly improved our model's performance (as shown in Table 5), the performance of hand-drawn sketches is still lower than that of computer-generated images. This difference reveals an important finding: synthetic data cannot completely replicate the unique characteristics of real hand-drawn sketches. We could only discover this insight by testing with actual hand-drawn data.
>
> ### 2. Applications Beyond Sketch-to-TikZ
> The value of SkeTikZ extends beyond sketch-to-TikZ conversion. Its paired data structure enables other applications, such as generating sketches from rendered images. Additionally, combining SkeTikZ with datasets from other programming languages could lead to more versatile models through multi-task learning. These possibilities demonstrate SkeTikZ's potential impact on both current and future research.
>
> > ### Are there any simple methods available for correction?
>
> One approach is to provide models with both generated outputs and code to help identify and fix errors. However, our early studies show that detecting and correcting subtle visual differences remains technically challenging.
> While automatic error detection and correction could improve accuracy, this remains an important area for future research.
> We will include this perspective in our manuscript as part of future work.
>
> > ### The method proposed in this article for converting hand-drawn sketches to vector graphics relies more on the combination of existing technical solutions.
>
> Following your helpful suggestions, we discuss our contributions from the following two perspectives.  We will enrich our manuscript with the points below to present a deeper discussion of our approach's contributions."
>
> ### 1. Discussion of multi-candidate Inference
>
> Our findings that multi-candidate inference proved highly effective for sketch-to-diagram generation provide important direction for future research in this field.
> While recent research [1,2] has demonstrated a similar approach of generating multiple candidates in text inference and coding tasks, we are the first to validate multi-candidate generation in sketch-to-diagram tasks. While common models like CLIP showed limited effectiveness, our specially designed D-SigLIP model proved highly effective at evaluating multiple generations, as shown in Figure 8. This result also highlights the importance of developing appropriate evaluation functions.
>
> #### [1] Brown et al., Large Language Monkeys: Scaling Inference Compute with Repeated Sampling, 2024
> #### [2] Snell et al., Scaling LLM Test-Time Compute Optimally Can be More Effective than Scaling Model Parameters, 2024
>
> ### 2. Discussion of data augmentation
>
> Conventional synthetic data generation for VLM training relied on image-understanding models to create question-answer pairs or descriptions from images. In contrast, our approach modifies TikZ code using text-based LLMs without the need for image comprehension. This enables us to generate numerous candidates at a significantly lower cost. Our study provides the first validation of this method. Since this approach is independent of TikZ code specifics, we believe it offers valuable insights for future research.

---

> ### Author Response · Authors · 2024-11-26
>
> We would like to thank you again for taking the time to review our paper. We have incorporated the following points into our revised paper. Due to space limitations, we have added a summary of the points above. For detailed discussions, please refer to our response above.
>
> > ### Whether the SKETIkZ dataset itself could play a significant role in future research endeavors.
>
> We have added a summary of the above discussion in lines 137 to 141 of Section 2 in the revised paper.
>
> >  ### The method proposed in this article for converting hand-drawn sketches to vector graphics relies more on the combination of existing technical solutions.
>
> We have added a summary of the above discussion in lines 252 to 255, 270 to 271, and 293 to 295 of Section 4 in the revised paper.

---

### Official Review · Reviewer_eF9v · 2024-11-03

**Soundness:** 2
**Presentation:** 3
**Contribution:** 2
**Rating:** 5
**Confidence:** 5

**Summary:**

This work investigates the problem of sketch-to-diagram generation, which converts hand-drawn sketches into diagrams formatted as TikZ codes. As drawing sketches is a visual tool that is more intuitive and user-friendly for ideation, the authors think tackling the problem of sketch-to-diagram generation is meaningful but underexplored so far, compared to its text-to-diagram generation counterpart. The authors proposed using a vision language model to handle the interesting problem. A new dataset of paired sketch images, TikZ codes and reference images is also proposed.

**Strengths:**

- The proposed dataset contains 3,231 sketch images associated with the TikZ codes, and the rendered images are valuable for sketch-to-diagram generation.
- It is interesting and reasonable that a vision language model is utilised to tackle the problem of sketch-to-diagram generation. This work expands the usage of VLM into a new domain.
- Extensive experiments are conducted, which is helpful in understanding the effectiveness and the limitations of the proposed method and other VLM models in the context of image-to-TikZ generation.

**Weaknesses:**

- The authors claim that the task of sketch-to-diagram has not been explored before. However, there are some existing works [a][b][c][d]. It would be nice if the authors discuss how this work differs from or improves upon these existing works, particularly in the context of TikZ code generation from sketches. This would help clarify the novelty and contributions of this work.
- The authors could further improve the dataset section. For example, the authors could demonstrate all the types of diagrams this work focuses on and provide statistics about the data (e.g., how many for each category, etc).
- The contribution of using data augmentation and multi-candidate inference tricks is minor.

[a] Vitruvion: A Generative Model of Parametric CAD Sketches, ICLR 2022
[b] Learning to infer graphics programs from hand-drawn images, NeurIPS 2018
[c] Neurosymbolic Models for Computer Graphics, Computer Graphics Forum 2023
[d] SketchGen: Generating Constrained CAD Sketches, NeurIPS 2021

**Questions:**

- Could the authors provide a detailed analysis of why the proposed method requires generating multiple candidates for the TikZ code and selecting the best one? Does this stem from the limitation of the vision encoder or the LLM-based code generator?
- Follow-up question: It seems the output of the CodeLLM is somehow random, i.e., sometimes it works and sometimes not, so it requires generating until a satisfactory result is given (both the iterative and the multi-candidate generation falls in this case). Is there any strategy to improve the consistency and accuracy of the code generation process?
- It is confusing to know the difference between CSR_avg and CSR_cum. How could N_gen be different from N_test? The authors might want to provide a concrete example that illustrates how these two metrics are calculated and why they might differ. This could help readers better understand and interpret the results.
- Regarding all the competitors, did all of them fine-tune using the SkeTikZ dataset as well?

---

> ### Author Response · Authors · 2024-11-19
>
> Thank you for your valuable comments. We appreciate your positive comments regarding the value of the dataset and the experimental results.  We hope our responses adequately address the questions and concerns raised.
>
> > ### It would be nice if the authors discuss how this work differs from or improves upon these existing works, particularly in the context of TikZ code generation from sketches.
>
> Thank you for your valuable comments. Studies [a] and [b] are related to our work addressing sketch image inputs.
> The following section details how our research differs from prior research. We will incorporate this point into our related work section.
>
> ### 1. Diagram Complexity and Scope
>
> #### While study [b] focuses on TikZ code generation for basic geometric shapes, our research addresses complex real-world diagrams from arXiv, including sophisticated geometric structures and textual elements such as node labels and mathematical expressions. Moreover, our approach handles sketches from various drawing tools, accommodating diverse noise characteristics.
>
> ### 2. End-to-end modeling
>
> #### Our end-to-end image-to-code model advances beyond conventional two-stage approaches [a,b] by enabling direct code generation without separate element identification stages. This unified approach facilitates efficient data collection while providing language-agnostic extensibility through multi-task learning.
>
>
> #### [c] serves as a survey paper of the previously discussed work. [d] addresses CAD program generation from elements and constraints rather than image-to-CAD conversion.
>
> > ### The authors could demonstrate all the types of diagrams this work focuses on and provide statistics about the data
>
> We included the statistics of diagram types in our dataset in Table 9 and examples of each category in Figure 9 of appendix section E due to space constraints. While our main text currently only references section E, we will revise the manuscript to explicitly include the reference to Table 9 and Figure 9.
>
> > ### The contribution of using data augmentation and multi-candidate inference tricks is minor.
>
> Following your helpful suggestions, we discuss our contributions from the following two perspectives. We will enrich our manuscript with the points below to discuss our approach's contributions more deeply.
>
> ### 1. Discussion of multi-candidate Inference
>
> #### Our findings that multi-candidate inference proved highly effective for sketch-to-diagram generation provide important direction for future research in this field.
> While recent research [1,2] has demonstrated a similar approach of generating multiple candidates in text inference and coding tasks, we are the first to validate multi-candidate generation in sketch-to-diagram tasks. While common models like CLIP showed limited effectiveness, our specially designed D-SigLIP model proved highly effective at evaluating multiple generations, as shown in Figure 8. This result also highlights the importance of developing appropriate evaluation functions.
>
> #### [1] Brown et al., Large Language Monkeys: Scaling Inference Compute with Repeated Sampling, 2024
> #### [2] Snell et al., Scaling LLM Test-Time Compute Optimally Can be More Effective than Scaling Model Parameters, 2024
>
> ### 2. Discussion of data augmentation
>
> #### Conventional synthetic data generation for VLM training relied on image-understanding models to create question-answer pairs or descriptions from images. In contrast, our approach modifies TikZ code using text-based LLMs without the need for image comprehension. This enables us to generate numerous candidates at a significantly lower cost. Our study provides the first validation of this method. Since this approach is independent of TikZ code specifics, we believe it offers valuable insights for future research.
>
> > ### Is there any strategy to improve the consistency and accuracy of the code generation process?
>
> Related to our response to your question "Why does the proposed method require generating multiple candidates for the TikZ code and selecting the best one?", we hypothesize that one key reason why the multiple-candidate approach proved effective is that code correctness and diagram image quality metrics are not incorporated during the training phase. We consider it an important future research direction to develop models that can generate diagrams with fewer errors and better visual quality by incorporating these metrics into the training process. We appreciate this insight and will include it in our future work discussion.

---

> ### Author Response · Authors · 2024-11-19
>
> > ### Difference between $CSR_{\text{avg}}$ and $CSR_{\text{cum}}$
>
> Following your valuable suggestions, we have provided detailed explanations below. We will incorporate these additional details into the appendix section to enhance the overall quality of our manuscript.
>
> Let us clarify the difference between $CSR_{\text{avg}}$ and $CSR_{\text{cum}}$.
>
> $CSR_{\text{avg}}$ represents the success rate across all generation attempts.
> For example, if a model attempts $N$ generation for each of the 100 test samples and succeeds in compilation $K$ times, then
> $$
> CSR_{\text{avg}} = \frac{N_{\text{success}}}{N_{\text{gen}}} = \frac{K}{(100 \times N)}
> $$
> To illustrate, if we make 10 generation attempts for each of the 100 test samples (totaling 1,000 generations) and achieve successful compilation in 400 cases, then
> $$
> CSR_{\text{avg}} = \frac{400}{1000} = 0.4.
> $$
>
> $CSR_{\text{cum}}$, which is exclusively used for iterative generation, measures the cumulative proportion of test samples achieving successful compilation across multiple attempts. Consider the following sequential process:
>
>  - first generation: 50 of the 100 samples compile successfully
>  - Second generation: 20 of the remaining 50 (100 - 50) samples compile successfully
>  - Third generation: 10 of the remaining 30 (50 - 20) samples compile successfully
>
> In this scenario,
> $$
> CSR_{\text{cum}} = \frac{N_\text{test success}}{N_\text{test}} = \frac{50 + 20 + 10}{100} = 0.8
> $$
> This metric specifically quantifies the proportion of test samples that eventually achieve successful compilation, independent of the total generation attempts.
>
> The motivation for utilizing these two distinct evaluation metrics arises from their complementary analytical perspectives:
> CSR_avg represents the average compilation success rate, enabling fair model comparison. CSR_cum measures the proportion of successfully compiled test samples across multiple attempts, analogous to a recall metric.
>
>
> > ### Did all of the competitors fine-tune using the SkeTikZ dataset as well?
>
> We evaluated four state-of-the-art AI models (Claude 3.5, GPT-4o, GPT-4o mini, and LLaVA-next) without SketikZ fine-tuning. We aimed to assess their natural ability to handle this task without specialized training.
>
> In our early research, we also tested closed models by giving them few-shot examples taken randomly from the SketikZ training data. However, we observed no significant improvement in performance. Additionally, we confirmed that simply fine-tuning conventional VLMs (like LLaVA 1.5) on SketikZ yielded poor performance. Given these preliminary findings, we have decided to focus on reporting the models' baseline performance in our current manuscript. We will include these initial experimental results in the final version of our paper. We would greatly appreciate any suggestions regarding additional experiments that might strengthen our paper.

---

> ### Author Response · Authors · 2024-11-25
>
> > ### Why does the proposed method require generating multiple candidates for the TikZ code and selecting the best one?
>
> Thank you for your insightful question. We would like to provide a strengthened response to your question and offer a detailed explanation as follows:
>
> ### 1. Characteristics of Code Generation
> Thank you for your insightful question. Code generation presents unique challenges distinct from text generation, as syntax variations can affect compilation and visual outputs. With reference code averaging 739 tokens, generating error-free code in a single attempt is particularly challenging for autoregressive models, where errors persist throughout generation. Through multiple candidate generation, we can identify and select outputs that minimize the occurrence of such errors.
>
> ### 2. Consideration of Image Similarity
> While our model was instruction-tuned for code generation using next-token prediction loss, the visual similarity of the compiled images should also be considered to generate high-quality candidates. Through multi-candidate generation, we significantly improved performance by incorporating this visual similarity metric in the post-processing stage.
>
> With this additional discussion, we will enhance our manuscript to emphasize the significance of multi-candidate generation.

---

> ### Author Response · Authors · 2024-11-26
>
> We would like to thank you again for taking the time to review our paper. We have incorporated the following points into our revised paper. Due to space limitations, we have added a summary of the points above. For detailed discussions, please refer to our response above.
>
> > ### It would be nice if the authors discuss how this work differs from or improves upon these existing works, particularly in the context of TikZ code generation from sketches.
>
> We have added a summary of the above discussion in lines 123 to 129  and 142 to 145 of Section 2 in the revised paper.
>
> > ### Why does the proposed method require generating multiple candidates for the TikZ code and selecting the best one?
>
> We have added a summary of the above discussion in lines 288 to 293 of Section 4 in the revised paper.
>
> > ### Is there any strategy to improve the consistency and accuracy of the code generation process?
>
> We have added a summary of the above discussion in lines 537 to 540 of Section 9 in the revised paper.
>
> > ### The authors could demonstrate all the types of diagrams this work focuses on and provide statistics about the data
>
> We have added a reference to the appendix E in lines 188 to 189 of Section 3 in the revised paper.
>
> > ### Difference between $CSR_{\text{avg}}$ and $CSR_{\text{cum}}$
>
> We have added an explanation in Appendix J in the revised paper.
>
> > ### The contribution of using data augmentation and multi-candidate inference tricks is minor.
>
> We have added a summary of the above discussion in lines 252 to 255, 270 to 271,  and 293 to 295 of Section 4 in the revised paper.

---

### Official Review · Reviewer_CrUP · 2024-11-03

**Soundness:** 2
**Presentation:** 1
**Contribution:** 2
**Rating:** 6
**Confidence:** 5

**Summary:**

This paper generates TikZ code and render them as images from input hand-drawn sketch and a textual prompt (predefined instruction). The proposed VLM, ImgTikZ, combines three components: an open-source LLM from DeepSeek coder, a vision encoder from SigLIP, a trainable adapter network (i.e., a linear layer) added to a pre-trained SigLIP vision encoder which is trained using contrastive learning, and a LoRA appended to the language model. The resulting method is trained in two-stages: in stage-1, the adapter network weights are updated, whereas, in stage-2, both the adapter and LoRA are updated. Along with the ImgTikZ method, the authors proposed SkeTikZ, a new dataset comprising of 3,231 pairs of hand-drawn sketches and corresponding TikZ codes. The authors further augment this dataset such as synthesising notebook backgrounds, adding Gaussian noise, varying brightness and contrast, and introducing distortion. The proposed method and the impact of the dataset is measured using four automatic evaluation: compilation success rate, image similarity, code similarity, and character similarity.

**Strengths:**

[+] I really want to appreciate the authors for highlighting concurrent works (Belouadi et al, 2024). This is something that should be celebrated more broadly, as it really helps readers understand the overall literature.

[+] The proposed method is simple and intuitive, without unnecessary forced contributions.

[+] The SkeTikZ dataset will hugely help the community in <query>-to-technical diagram generation.

**Weaknesses:**

[-] While the proposed method is simple and clear, the paper, however, is difficult to follow. For example, when describing "model structure" in Sec.4.1 or in "Datasets used in stage 1 training" the authors could describe the simple adapter network. I could not find the architecture of this linear layer until I went to Page-15 in supplementary. The same is for LoRA, where the authors have to wait till Page-15 in Tab.7 to know lora_r and lora_alpha. If space is the limitation, the authors could add the following in Sec.4.1 -- "for more details on <xxx> architecture/designs, please refer to Tab.7".

[-] The dataset creation process has a caveat: images are rendered using pdflatex, after which a human annotator draws a sketch based on the rendered image. However, human sketching is inherently a lossy process, meaning that certain details may be ommitted or subtly altered. Consequently, when a VLM uses this dataset to perform sketch-to-image generation, there is a high risk of hallucination where the model might introduce small details into the generated image that were not present in the original sketch. Hallucinations may not always be a bad thing, but for generating code or technical diagrams, this can be detrimental.

[-] How is the proposed method different from Gervais et al., 2024 and others, which generate LaTeX code from screenshots of mathematical formulas or handwritten images? Can they be applied to this same problem with minimal modifications?

Minor:

[-] Typo I presume? re: CSR_ave in Tab.2 whereas CSR_avg in Tab.3

[-] All the tables look too big and consumes a lot of unnecessary space. I would suggest the authors to make the table font smaller and use the space to add details on adapter network, LoRA, and some training details.

**Questions:**

See weaknesses

---

> ### Author Response · Authors · 2024-11-19
>
> Thank you for your insightful comments. We appreciate your positive feedback regarding the dataset's value and our proposed method. We hope our responses adequately address the questions and concerns raised.
>
> > ### Location of Model Architecture Details
>
> As you pointed out, we had placed some model architecture details in the appendix due to space limitations. We will add the statement 'for more details on <xxx> architecture/designs, please refer to Tab.7' to Section 4.1. Additionally, we will compress the table and incorporate the description of linear layers into the main text.
>
> > ### On Information Loss in Hand-Drawn Representations
>
> We minimize such losses through careful dataset curation, instructing annotators to maintain fidelity to reference images, and excluding overly complex diagrams. Moreover, successful diagram generation often requires models to complement and enhance input sketch information—for instance, recognizing and completing discontinuous lines in hand-drawn geometric shapes. This information completion capability represents a crucial aspect of converting sketches into well-formed diagrams.
> Regarding the generation of diagrams that accurately reflect user intentions, we consider the development of an interactive refinement system to be a significant direction for future research. Such a system would enable users to modify outputs that deviate from their intended representations. We will include a discussion of this point in our final manuscript.
>
> > ### How is the proposed method different from Gervais et al., 2024 and others, which generate LaTeX code from screenshots of mathematical formulas or handwritten images?
>
> Thank you for your question. Our approach differs in the following three aspects. We will include the following detailed discussion in the related work section.
>
> ### 1. While Gervais et al. (2024) fine-tuned VLMs pre-trained for general image-to-text tasks, we developed a code-specific VLM by combining a code-LLM with a vision encoder to create a model specialized for code generation.
>
> #### Our preliminary studies revealed that code-specific LLMs significantly outperformed fine-tuned general-purpose VLMs for our task. This performance difference could be attributed to two key factors: first, the substantially longer token sequences required for diagram generation (739 tokens on average, compared to 65 tokens for the image-to-latex task [1] and 30 tokens for the conventional VLM [2]), and second, the need to understand complex two-dimensional spatial relationships between diagram elements, unlike simpler OCR-like tasks in mathematical formula generation. These challenges highlight the necessity of employing LLMs with specialized programming knowledge.
>
> ##### [1] Deng et al., Image-to-Markup Generation with Coarse-to-Fine Attention, ICML2017
> ##### [2] Liu et al., Visual Instruction Tuning, Neurips2023
>
> ### 2. We introduced a new approach using multiple candidates during inference, which significantly enhanced our model's performance.
>
> #### The complexity of generating long sequences made our proposed multi-candidate generation approach particularly effective. Previous image-to-latex research has not explored this effectiveness, and our study was the first to uncover this finding.
>
> ### 3. We demonstrated the effectiveness of two data augmentation strategies: code modification using GPT-3.5 to transform TikZ code segments and image augmentation applied to the diagram data.
>
> #### These data augmentation techniques have not been previously applied to mathematical formula generation tasks. They are particularly crucial in mapping between two-dimensional variations of diagrams and their respective code representations.

---

> > ### Comment · Reviewer_CrUP · 2024-12-02
> >
> > Thanks for the detailed responses and for incorporating these discussions into the main paper -- this certainly helped, and I'm inclined to increase my score.
> >
> > I still have one concern about the 'Information Loss in Hand-Drawn Representations' section. While you discuss how models need to handle discontinuous lines in sketches, I believe information loss extends beyond just broken lines. Hand-drawn diagrams often miss smaller details and elements that could subtly alter the diagram's intended meaning. I'm not entirely convinced the SKETIkZ dataset addresses these cases adequately. This might be worth exploring more thoroughly in your future work.

---

> ### Author Response · Authors · 2024-11-26
>
> We would like to thank you again for taking the time to review our paper. We have incorporated the following points into our revised paper. Due to space limitations, we have added a summary of the points above. For detailed discussions, please refer to our response above.
>
> > ### Location of Model Architecture Details
>
> We have added an explanation in lines 208 to 209 and  211 to 212 of Section 4 in the revised paper. We also provided the hyperparameter details in Section 6, lines 356-359.
>
> > ### On Information Loss in Hand-Drawn Representations
>
> We have added a summary of the above discussion in lines 535-537 of Section 9 in the revised paper.
>
> > ### How is the proposed method different from Gervais et al., 2024 and others, which generate LaTeX code from screenshots of mathematical formulas or handwritten images?
>
> We have added a summary of the above discussion in lines 101-106 of Section 2 in the revised paper.
>
> > ### Minor Issues
>
> We have revised the manuscript following your suggestions.

---

> ### Author Response · Authors · 2024-12-04
>
> Thank you for your constructive feedback. We appreciate your recognition of our dataset’s value in the strengths section. At the same time,  while we instruct annotators to create sketches that match the original diagrams as closely as possible, we acknowledge that annotators may inadvertently omit small elements from the original diagrams as you pointed out. While a straightforward approach to minimize these omissions would be having annotators create both sketches and TikZ code, this method would require significant resources for TikZ code creation. Our method of sketching rendered images provides a reasonable approach for creating thousands of examples. Following your advice, we plan to investigate this information loss aspect further in our future work.

---

### Official Review · Reviewer_jbiu · 2024-11-06

**Soundness:** 3
**Presentation:** 3
**Contribution:** 3
**Rating:** 8
**Confidence:** 4

**Summary:**

The paper presents a system for generating high-quality vector diagrams from hand-drawn sketches based on the new SKETIkZ dataset. SKETIkZ has pairs of hand-drawn sketches and TikZ codes. The system uses data augmentation and a multi-candidate inference strategy to significantly improve output quality.

**Strengths:**

- This paper proposes a novel important dataset. SKETIkZ fills a critical gap in publicly available data for sketch-to-diagram conversion, supporting future research.
- IMGTIkZ shows comparable performance to larger models like GPT-4o, highlighting efficient architecture and training strategies.
- Data augmentation and multi-candidate inference enhances performance.
- This paper contributes SKETIkZ for evaluating vision-language models' capabilities in diagram generation from sketches.
- Human evaluation is adopted to evaluate the generation results.

**Weaknesses:**

- No attempt was made to train with other open-source multimodal large models.
- No performance comparison was made with other open-source multimodal large models.
- Other input should be considered, such as a hand-drawn diagram with corresponding descriptions, where descriptions can be generated using a captioning model.

**Questions:**

Would it be better to add text descriptions as input along with the hand-drawn diagram for image input?

---

> ### Author Response · Authors · 2024-11-19
>
> Thank you for your valuable comments. We appreciate your positive feedback regarding the value of SkeTikZ and the effectiveness of our proposed methodology. The following response addresses the questions and concerns raised.
>
>
> > ### Experiments with Other Open-Source Vision-Language Models
>
> In our preliminary studies, we conducted two experiments. First, we tested Code Llama as an alternative to DeepSeek Coder. Second, we explored fine-tuning the original LLaVA-1.5. However, both approaches yielded lower performance compared to our proposed model. We will include these experimental results in the supplementary material. We would greatly appreciate any suggestions regarding additional experiments that might strengthen our paper.
>
> > ### Would it be better to add text descriptions as input along with the hand-drawn diagram for image input?
>
>
> Thank you for your suggestion. Our preliminary studies explored an approach using GPT-4V to generate intermediate textual descriptions of hand-drawn diagrams before code generation. However, this method did not significantly improve performance, likely due to two main factors: recognition errors in hand-drawn diagrams and increased ambiguity from text-based representations. These findings suggest that leveraging text generation does not offer a simple solution for enhancing model performance.

---

> > ### Comment · Reviewer_jbiu · 2024-11-26
> >
> > Thank you for the author's reply which addressed my most of concerns.
> > Could the authors share the preliminary study of textual description from GPT-4V as an input or include them in the final paper? This would be helpful for complementing this paper, also for readers to fully understand this work.

---

> ### Author Response · Authors · 2024-11-26
>
> Thank you very much for taking the time to provide us with valuable comments!
>
> > ### Could the authors share the preliminary study of textual description from GPT-4V as an input or include them in the final paper?
>
> Yes, we are currently in the process of incorporating this content into our manuscript and will include it in the camera-ready version.

---

### Author Response · Authors · 2024-11-26
**General Response to All Reviewers**

We sincerely thank the reviewers for taking the time to provide constructive and valuable feedback on our manuscript. We also appreciate their positive comments on our dataset, methodology, and experiments. We have uploaded the revised paper incorporating the discussion feedback, with new additions highlighted in blue. We believe these revisions have enhanced both the completeness and clarity of our manuscript. All specific changes made to the manuscript have been detailed in our responses to the reviewers. We would be grateful if we could hear your feedback regarding our responses to the reviews.

---

### Author Response · Authors · 2024-12-02
**Dear Reviewers,**

We would like to express our sincere gratitude to the reviewers for their thoughtful evaluation of our work. We have carefully addressed all review comments and updated our paper accordingly. We believe that the concerns raised have been satisfactorily resolved. As we approach the end of the discussion period, we welcome additional comments and questions if you have any remaining concerns. Thank you once again for your time, effort, and valuable insights.

---

### Meta-Review · Area_Chair_JYrK · 2024-12-22

**Metareview:**

This paper initially received 2 borderline reject and two positive reviews. Almost all reviewers felt the proposed SKETIkZ dataset will be useful for the research community.  Also the authors response help the reviewer CrUP, and the score was increased to borderline accept (6).  Reviewer jbiu also satisfied with the responses. Unfortunately other two reviewers did not participate in the discussion and left the final scores unaltered. This paper finally ended with three supporting reviews and a borderline reject. Considering the reviews and responses, the AC panel acknowledged the importance of the introduced dataset for research community and decided to accept the paper. Authors should update the manuscript considering all the reviewers suggestions while preparing the camera ready.

**Additional Comments On Reviewer Discussion:**

Authors provided responses for all the reviewers. Only two reviewers acknowledged the responses and participated in the discussion. Over all there was a positive sentiment about the paper and all the reviewers acknowledged the importance of the the proposed dataset. Post  rebuttal, the senior reviewer (CrUP) who works in this area increased the score from 5 to 6 indicating acceptance of the paper.

---

### Decision · Program_Chairs · 2025-01-22

Accept (Poster)